# Deep phenotypic characterization of immunization-induced antibacterial IgG repertoires in mice using a single-antibody bioassay

Millie Heo [1], Guilhem Chenon[1], Carlos Castrillon[2,3], Jérôme Bibette[1], Pierre Bruhns[2], Andrew D. Griffiths [3], Jean Baudry [1] & Klaus Eyer [1,4 ✉]

Antibodies with antibacterial activity need to bind to the bacterial surface with affinity, specificity, and sufficient density to induce efficient elimination. To characterize the anti-bacterial antibody repertoire, we developed an in-droplet bioassay with single-antibody resolution. The assay not only allowed us to identify whether the secreted antibodies recognized a bacterial surface antigen, but also to estimate the apparent dissociation constant ($K_{D\ app}$) of the interaction and the density of the recognized epitope on the bacteria. Herein, we found substantial differences within the $K_{D\ app}$/epitope density profiles in mice immunized with various species of heat-killed bacteria. The experiments further revealed a high cross-reactivity of the secreted IgG repertoires, binding to even unrelated bacteria with high affinity. This application confirmed the ability to quantify the anti-bacterial antibody repertoire and the utility of the developed bioassay to study the interplay between bacteria and the humoral response.

[1] 'Laboratoire Colloïdes et Matériaux Divisés' (LCMD), ESPCI Paris, PSL Research University, CNRS UMR8231 Chimie Biologie Innovation, F-75005 Paris, France. [2] Unit of Antibodies in Therapy and Pathology, Institute Pasteur, UMR1222 INSERM, F-75015 Paris, France. [3] 'Laboratoire de Biochimie' (LBC), ESPCI Paris, PSL Research University, CNRS UMR8231 Chimie Biologie Innovation, F-75005 Paris, France. [4] Laboratory for Functional Immune Repertoire Analysis, Institute of Pharmaceutical Sciences, D-CHAB, ETH Zürich, Zürich, Switzerland. ✉email: Klaus.eyer@pharma.ethz.ch

Antibodies are important mediators to overcome bacterial infections[1,2]. To do so, these antibodies not only need to bind to bacteria, but also have to induce a specific response within the host organism to allow for bacterial elimination[3,4]. Not every antibody that recognizes a bacterial surface antigen is able to induce such a response. First, only a few specific antibody isotypes are able to interact with antibody receptors on immune cells and complement proteins, and therefore to trigger secondary reactions[5–8]. Second, the antibody needs to strongly attach itself onto the bacterium, and therefore also the binding strength is important[9]. Indeed, the efficiency of bacterial killing through opsonophagocytosis has been shown to correlate strongly with affinity, but no correlation has been observed when antibodies binding to different antigens were compared[8]. Therefore, the epitope availability and density on the bacterial surface is a third key parameter when studying the capability of antibodies to eliminate bacteria. Indeed, the bacterial membrane contains many different potential antigens, present in various concentrations. About $3 \times 10^5$ lipopolysaccharide (LPS) molecules and $1 \times 10^5$ various proteins per $\mu m^2$ were described in *Salmonella* outer membrane for example[10–12], and many of these lipids, proteins, and sugars are potential antigens recognized by antibodies. Many of these antigens also contain more than one epitope, i.e., potential antibody binding sites[13,14], indicating an enormous number and variety of potential epitopes. Additionally, many of the bacterial antigens are species-specific but share structural domains and similar chemical compositions[15–17], leading to potentially cross-reactive antibodies. In the past, the study and characterization of anti-bacterial antibodies generated upon immunization was assayed by measuring the presence of antibodies within the serum. These assays have been recently re-developed to add functionalities such as bacterial killing[18–20]. Indeed, systems serology characterized the integral of the antibody response and its functionalities, but due to the complex composition of the serum the analytical resolution into the repertoire (i.e., the diverse pool of present antibodies) was limited[21,22].

For proper biophysical and functional characterization, antibodies are preferably assayed as monoclonal species[23,24]. This demand for single-antibody characterization can be ultimately linked to single-cell resolution since each cell is only capable of producing one antibody variant at a given time[25]. Therefore, cell-based technologies such as flow cytometry[26], enzyme-linked immunospot assays (ELISPOT)[27] and droplet-microfluidics[28–30] have been used to study anti-bacterial antibody repertoires. All of these technologies allow to screen the secreted antibody repertoire for surface-binding antibodies with high-throughput on the single-antibody and cell level; however, they offer low analytical

resolution in terms of affinity and epitope density. Recently, we demonstrated an in-droplet phenotypic immunofluorescent bioassay to quantitatively describe the IgG-repertoire with single-antibody resolution[31]. In this technology, fluorescence relocation immunoassays with single-antibody resolution were performed to not only detect IgG, but to determine secretion rates and to calculate the affinity of said IgG against a defined, soluble antigen. The antibody and antigen relocation was measured on magnetic nanoparticles that formed an observable object (beadline) within each droplet; allowing to standardize the assay, and the measured antibodies were secreted by individual, co-encapsulated antibody-secreting cells (ASC) extracted from immunized mice. Although the previous bioassay was able to characterize the secreted antibody repertoire with high analytical resolution, such as individual secretion rates and affinities, the range of usable antigens was limited to purified, soluble proteins.

In the study presented herein, we were interested in quantifying and characterizing the immunization-induced anti-bacterial IgG repertoire that was able to interact with bacterial surface antigens; and to do so with single-antibody resolution. We demonstrate the use of an adapted version of our in-droplet bioassay that enabled the screening and characterization of the secreted antibody repertoire with single-antibody resolution; and the assays' capacity to extract affinity and epitope density for each individual antibody. Next, we immunized mice with various heat-killed bacteria and characterized the immunization-induced anti-bacterial IgG repertoire; their individual apparent affinities against the bacteria used for immunization but also their cross-reactive potential against a variety of other bacteria. To conclude, our bioassay allowed the classification and comparison of specific and cross-reactive anti-bacterial antibodies according to their apparent dissociation constant ($K_{D\ app}$) and epitope availability, and revealed highly cross-reactive IgG repertoires following immunization with full heat-killed bacteria, binding to even unrelated bacteria with high affinity.

## Results

**Bactoline development and initial characterization.** To comprehensibly address the diversity of surface bacterial antigens, we directly covered the nanoparticles with heat-killed bacteria (HK-B) (Figs. 1, 2a; "Methods"). The bacterial cells were immobilized on 300 nm streptavidin paramagnetic nanoparticles via a biotinylated version of a hydrophobic cholesterol linker[32], in which the cholesterol moiety interacted with the hydrophobic domains on the bacterial cell surface and the biotin with the streptavidin present on the nanoparticle surface (see also "Methods"). The bacteria-covered nanoparticles, termed bactoline, served as the

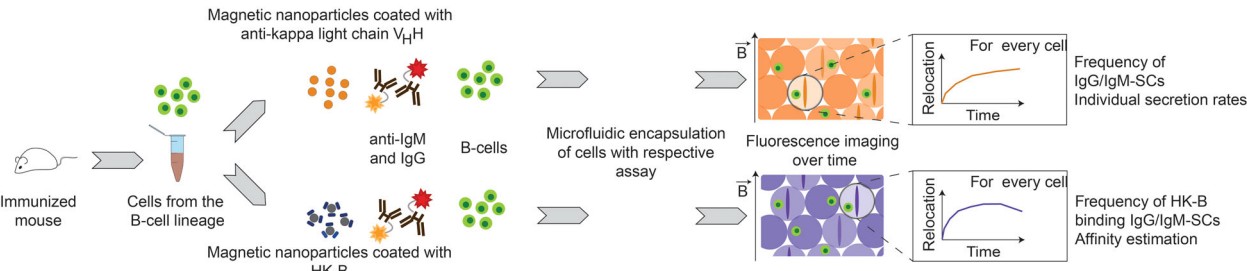

**Fig. 1 Overview of the droplet experiments employed in this study.** In this study, we extracted cells from the B-cell lineage from immunized mice, and encapsulated these cells either with a bioassay that allows to measure antibody secretion and the total frequency of IgG- and IgM-SCs (beadline assay, regardless of recognized antigen, top), or a bioassay to specifically measure anti-bacterial antibodies (bactoline assay, bottom). Individual cells are encapsulated with the respective assays, the droplets mapped in two-dimensions and antibody secretion and binding is measured over time. The single-cell and single-antibody data from both assays is combined afterwards, allowing to measure frequencies of total and specific IgG/IgM-SCs, individual secretion rates, and finally to estimate affinities and epitope densities.

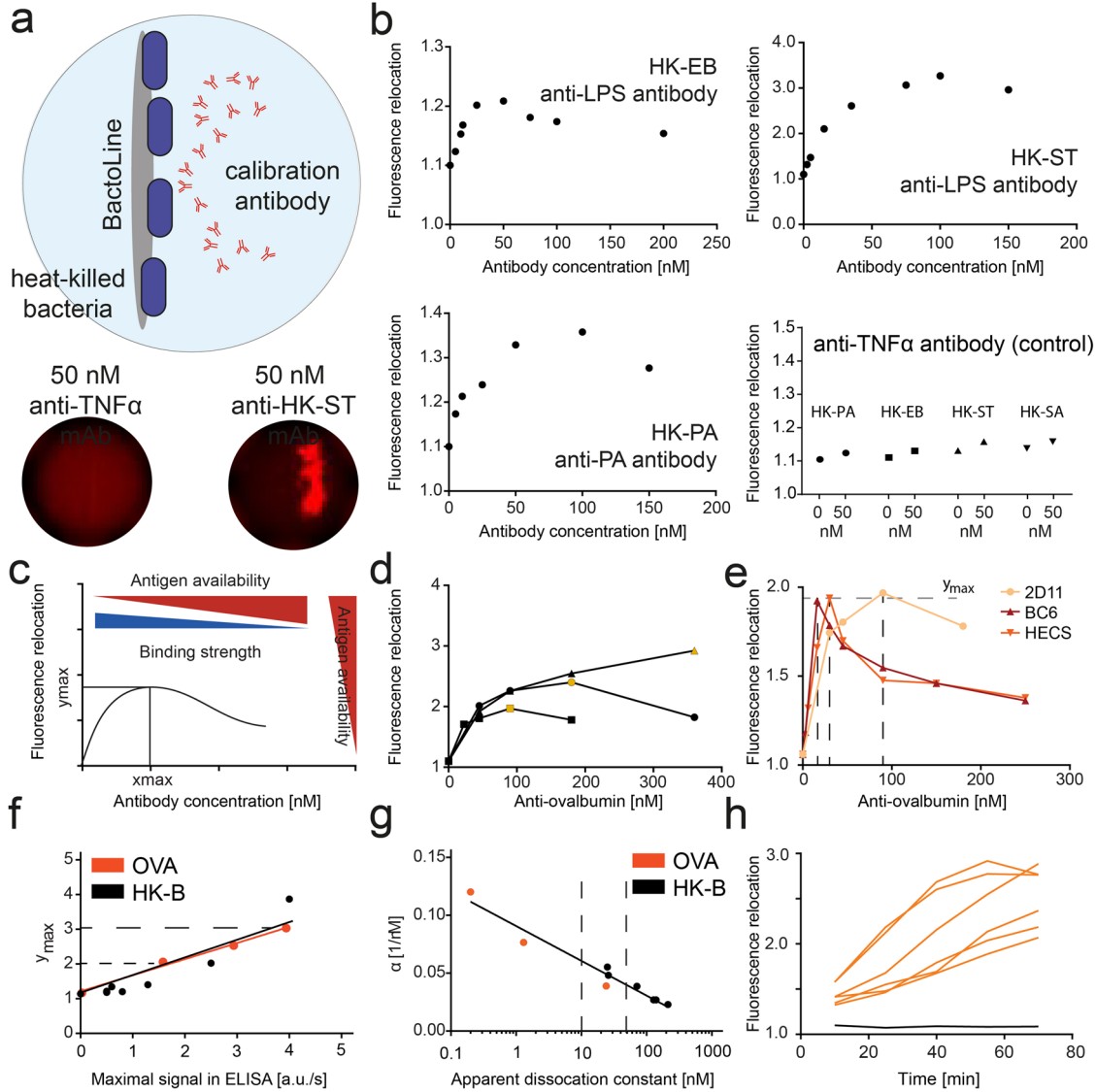

**Fig. 2 Bactoline characterization using commercial antibodies and model antigens. a** Scheme and micrographs of bactolines. For bactoline measurements, the HK-B were directly immobilized on the magnetic nanoparticles. By applying a magnetic field, the nanoparticles formed a solid surface used for fluorescence relocation immunoassays. The micrographs show droplets with bactolines (covered with HK-ST) that were generated in presence of 50 nM anti-TNFα (no relocation onto HK-B) or 50 nM anti-*S. typhimurium* LPS antibody (relocation visible). **b** Measured fluorescence relocation for HK-EB bactolines with a commercial anti-*E. coli* LPS antibody, for HK-ST with an anti-*S. typhimurium* LPS antibody, for HK-PA with an anti-*P. aeruginosa* antibody and an unrelated control antibody assayed with all three later used HK-B (anti-TNFα antibody), as well as HK-SA (*S. aureus*) for comparison. **c** Scheme describing the influence of binding strength and epitope density on the shape of the fluorescence relocation curves (**b**), especially on the position of the maximal fluorescence ($y_{max}$, $x_{max}$). **d** Measured fluorescence relocation for anti-OVA antibody 2D11 on OVA-covered nanoparticles. Shown are three curves with in-droplet concentrations of 10 (squares), 20 (circles) and 100 nM OVA (triangles). Shown in orange are the respective maximal relocation values. **e** Measured fluorescence relocation for different commercially available anti-OVA antibodies with an in-droplet OVA concentration of 10 nM. $K_{D\ app}$ varied, as measured by ELISA, from 0.2 nM (BC6), to 1.5 nM (HECS) and 24 nM (2D11). **f** Extracted $y_{max}$ values correlated strongly with the maximal binding observed in ELISA for OVA (red) and HK-B (black). **g** α; i.e., $x_{max}/y_{max}$, versus measured $K_{D\ app}$ from ELISA (see "Methods") for OVA (red) and HK-B (black). This curve was used for the estimation of $K_{D\ app}$ in each individual droplet. **h** Binding curves of murine IgG-SCs extracted from spleens of OVA-immunized mice, measured against OVA-coated beadlines. Shown are six example curves for cells secreting IgG that bound OVA (orange), and six cells that did not secrete IgG (black). In this figure, all droplet data is shown as mean values with SEM ($n > 3000$ droplets for each point, unless stated otherwise).

active surface to perform in-droplet immunoassays to detect bacteria-binding antibodies (Supplementary Fig. 1). The direct immobilization using the hydrophobic cholesterol linker minimized epitope masking and allowed to present the antigens in their natural environment, the bacterial membrane. Therefore, all accessible surface epitopes present during immunization were also accessible in the analysis. Additionally, the immobilization

via this linker was found to be stable over the time needed to perform the assays.

To reduce the impact of cellular heterogeneity[33–35] and to standardize the total amount of available antigens, we aimed to encapsulate at least a few hundred bacteria per droplet. The autofluorescence of HK-B[36] conveniently allowed to visualize the area of the bactoline, and to estimate the number of bacteria in the

area (164 ± 30 HK-B per droplet). Since this calculation assumed the bacteria to be only present in a 2-dimensional sheet rather than a 3-dimensional bactoline, this number represented a lower limit of the number of bacteria that are present within a droplet. Indeed, by measuring the 3D-volume of the bactolines by microscopy and assuming a volume of 2 μm³ per HK-B[37], the maximum number of bacterial cells in the bactoline volume was calculated to be around 1000 HK-B. This upper limit of HK-B per droplet was confirmed by the measurement of a 50-fold decrease in $OD_{600}$ before and after the immobilization step, corresponding to roughly 800 HK-B per droplet. Therefore, we estimated that between 200 and 800 HK-B were present within each droplet. This number was deemed high enough to allow for a homogenous presence of antigens in every droplet (coefficient of variance 3–18%, depending on method for calculation). Therefore, the influence of bacterial heterogeneity was reduced while still providing a well-standardized and homogenous surface for immunoassays. Herein, we successfully used this protocol to immobilize three different Gram-negative and a Gram-positive HK-B (either *P. aeruginosa* (PA), *E. coli O111:B4* (EB), *S. typhimurium* (ST), and *S. aureus* (SA)).

**Bactoline characterization—extraction of binding strength and epitope availability**. Next, we used commercially available monoclonal IgGs to characterize the fluorescence relocation profiles in response to a varying antibody concentration (Fig. 2, for the list of antibodies please refer to Supplementary Table 1). As expected, the addition of anti-bacterial antibodies resulted in a concentration dependent relocation of the fluorescent anti-Ig probe to the bactoline, similar to the curves shown in other homogenous immunoassays[31,38]. Unrelated antibodies, such as antibodies against TNFα, resulted in negligible increase in fluorescence for all the assayed HK-B (Fig. 2b). The overall shape of the curves, increasing fluorescence relocation followed by a decrease, are typical for homogenous immunoassays displaying a Hook effect[31,38,39]. Above the capacity of the bacteria for the antibody, any additional molecules will compete with the immobilized antibodies for the detection antibodies, therefore resulting in a decrease in fluorescence relocation. Interestingly, although the general shape was similar for all three HK-B and antibodies assayed (Fig. 2b), clear differences were visible in the curves. The different bacteria-specific commercial IgGs showed a wide range of relocation maxima ($y_{max}$, Fig. 2b). Maximal bactoline fluorescence relocation by anti-ST IgG was as high as 3.4, whereas $y_{max}$ in the HK-EB experiment was only 1.2. Furthermore, the position of said maxima along the x-axis, $x_{max}$, varied widely. For anti-ST IgG and anti-PA IgG, fluorescence relocation was found to increase up to around 100 nM of IgG, while J5 anti-LPS IgG showed a maximum peak at around 50 nM. Thus, we observed different $y_{max}$ and $x_{max}$ for different calibration antibodies.

We assumed that these differences were due to various bactoline capacities (i.e., different number of available epitopes) and antibody affinities (Fig. 2c). Indeed, the observed maximal fluorescence relocation ($y_{max}$) was modeled to be dependent on the available epitopes (higher relocation with more available epitopes, see Supplementary Information). Within the same model, the concentration of antibody at which the maxima were observed ($x_{max}$) was dependent on the available epitopes as well as the apparent dissociation constant ($K_{D\ app}$) of the antibody (see Supplementary Information and Supplementary Figs. 1–3 for modeling results). To confirm these modeling results in our experiments, we covered the nanoparticles with a simpler model antigen, biotinylated ovalbumin (OVA) instead of bacterial cells. Here, the concentration of immobilized antigen was controllable

and known; and therefore also the maximum capacity of the nanoparticles within each droplet. First, we varied the amount of antigen per droplet and observed the fluorescence relocation at different monoclonal anti-OVA antibody concentrations (Fig. 2d). Indeed, a controlled decrease in available epitopes resulted in a decrease of the observed $y_{max}$ as well as $x_{max}$ (Fig. 2d, orange markers). The nanoparticles were saturated with OVA at a concentration of 100 nM in the used dilution of particles[31], and consequently further increases in added antigen during incubation did not increase $y_{max}$ significantly.

Next, we fixed the antigen to 10 nM in-droplet concentration but used three monoclonal anti-OVA antibodies with different affinities (as estimated by ELISA $K_{D\ app}$ 0.2, 1.5 and 24 nM) to study the influence of binding strength on the resulting curves. In these experiments, we varied the concentration of each individually added monoclonal anti-OVA antibodies and measured the resulting fluorescence relocation (Fig. 2e). As expected, the variation of affinity resulted in different fluorescence relocation values at identical concentrations, but the overall shape of the curves and the $y_{max}$ value was conserved (all curves reach a maximum of fluorescence relocation around 1.91–1.96). This was also shown in the model (Supplementary Figs. 2, 3), where changes in affinity did not substantially influence $y_{max}$. Therefore, $y_{max}$ correlated directly with the concentration of available antigen, but was not influenced considerably by the affinity of antibodies (Fig. 2d, e). Furthermore, $y_{max}$ correlated nicely with total number of available epitopes found in ELISA for both OVA and HK-B assays (Fig. 2f, slope 0.46 ± 0.03; $R^2$ 0.99 for OVA measures, and slope 0.67 ± 0.08, $R^2$ 0.90 for HK-B), as measured as the maximum turnover rate. To conclude, $y_{max}$ was consequently used to determine the in-droplet concentration of available epitopes according to the findings displayed in Fig. 2f.

The second parameter, $x_{max}$, was influenced by affinity as well as epitope density (Fig. 2d, e). Since $y_{max}$ correlated only with epitope density, we used this parameter to correct $x_{max}$ for the influence of epitope density. This corrected factor, α, correlated nicely with the apparent dissociation constant ($K_{D\ app}$) for the simplified OVA system as well as the bactoline assay and its specific antibodies (Fig. 2f, g). Lastly, we combined the OVA assay with murine splenocytes from an OVA-immunized mouse to provide evidence that immobilized antigen allowed to measure antigen-specific antibodies. Here, we extracted murine ASCs from the spleens of OVA-immunized mice and measured the ability of the secreted antibodies to bind to the antigen that was immobilized on the beadlines. Indeed, the developed direct fluorescence relocation assay was able to visualize antibody binding to the antigen when primary cells were used (Fig. 2h).

**Immunizations with different HK-B lead to specific, but distinct antibody immune responses**. Next, we applied the bactoline assay to analyze the humoral immune responses ex vivo after immunization with bacteria. Mice were immunized with 10⁸ of each HK-B (either *P. aeruginosa* (PA), *E. coli O111:B4* (EB), *S. typhimurium* (ST)) in alum, or alum alone as a negative control (Fig. 3a). Cells from the B-cell lineage were purified 6 days after secondary immunization from the spleen, and beadline[31] and bactoline assays were performed in parallel. The beadline measurements (nanoparticles covered with anti-mouse kappa light chain V_HH) were used to determine the total frequency of immunoglobin G- and M-secreting cells (IgG-SCs and IgM-SCs), secretion rates[31] and the ability of the immunizations to induce class-switching to IgG; whereas the bactoline assay was used to measure the frequency and characteristics of bacteria-specific antibodies. The employed fluorescence detection assay for IgG

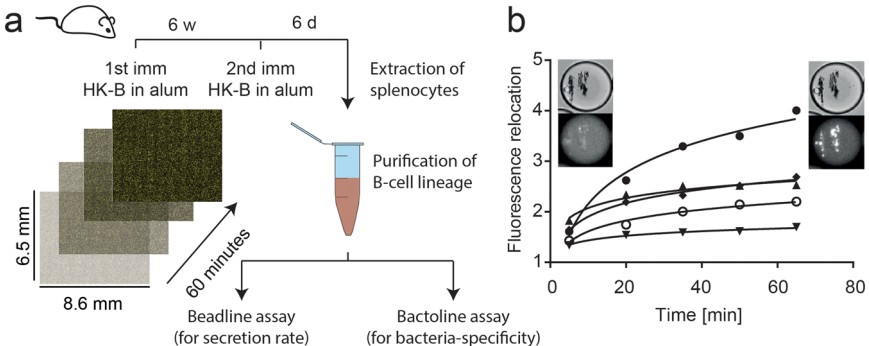

**Fig. 3 Overview of the animal experiments and the employed bactoline assay. a** Mice were immunized with $10^8$ HK-B in alum (or alum alone); and after 6 weeks boosted with the same immunization. Six days afterward, spleens were extracted and cells from the B-cell lineage were purified. Cells were split, and half of them were measured using a beadline assay (for measurement of secretion rates and frequencies of IgG- and IgM-SCs) and the other half in bactoline assays to measure bacteria-binding IgG. **b** Example curves of individual cells from a bactoline assay. Shown are fluorescence relocation values extracted from five different individual cells that secreted an IgG binding to HK-B. The micrograph shows the bright field and corresponding fluorescence image of a IgG-SC (full circle data) at time 0 and 60 min.

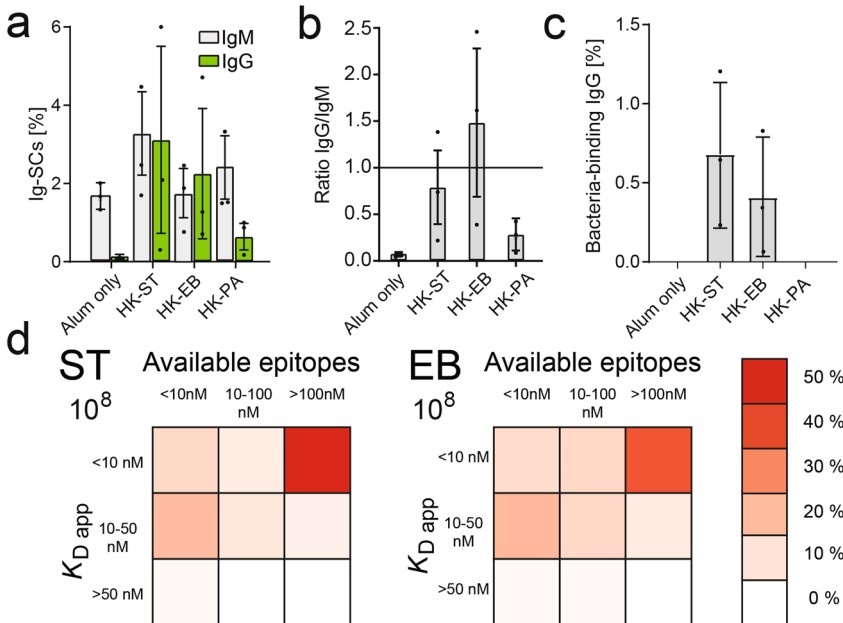

**Fig. 4 Immunization scheme and raw data of HK-B immunized mice. a** Frequency of IgM- and IgG-SCs in mice immunized with alum alone or different HK-B. **b** Ratio of class-switched IgG over IgM-SCs. **c** Frequency of bacteria-binding IgG-SCs in the different immunizations. **d** Characterization of HK-B binding IgG in HK-ST immunized (left, and HK-EB immunized mice (right). Data are cumulated from all three immunized mice (for each panel $200 > n_{anti-HK-B\ IgG-SCs} > 50$), binned as described in the methods and shown as a relative frequency of IgG in bin over total affine IgG-SCs. All frequency data (**a**–**c**) in this figure is shown as the mean frequency ± standard deviation over all mice ($n = 3$).

was able to detect the three most dominant isotypes IgG1, IgG2a, and 2b[31], whereas the IgM reporter was specific for IgM.

We observed a variable frequency of total IgM-SCs and IgG-SCs in mice immunized with different HK-B. The frequencies of total Ig-SCs, the sum of IgG-SCs and IgM-SCs, were $3.3 ± 1.0\%$ for the HK-PA immunized mice, $3.8 ± 2.1\%$ for the HK-EB immunized mice and $6.4 ± 3.8\%$ for the HK-ST immunized mice. Control mice immunized with adjuvant alone (alum) showed significant lower frequencies of $1.8 ± 0.4\%$ Ig-SCs ($p$-value for all <0.05, all indicated $p$-values are calculated using two-tailed Students $t$-tests). Thus, immunization with the HK-B increased the frequencies of the total ASCs up to 3-fold; but in a varying degree for the different HK-B. The increase in total Ig-SCs in HK-ST immunized mice was also reflected by a significant increase in IgM-SCs ($3.2 ± 1.2\%$; $p$-value = 0.01, Fig. 4a); whereas immunizations with HK-EB and HK-PA did not result in significantly

higher frequencies of IgM-SCs when compared to control mice immunized with alum. As expected, we observed increased frequencies of class-switched IgG-SCs for all HK-B immunizations compared to alum-only control mice ($p$-value for all < 0.01). As expected, adjuvant alone control mice showed almost exclusively IgM-SCs ($1.6 ± 0.3\%$ IgM-SCs; compared to $0.15 ± 0.04\%$ IgG-SCs, Fig. 4a). This was visualized as a class-switching ratio (frequency of IgG-SCs divided by the frequency of IgM-SCs, Fig. 4b). This ratio was around 1.0 for immunization with HK-ST and HK-EB (significantly different from alum control mice, $p$-value < 0.01, Fig. 4b). Interestingly, although HK-PA immunized mice showed only a small but significant increase in IgG-SCs, their class-switching ratio remained low. Cellular secretion rates for IgG- and IgM-SCs were measured in parallel and we found no significant difference in secretion for IgG- and IgM-SCs in individual immunizations with the different HK-B (Supplementary

Figs. 4 and 5, Median IgG secretion rate $250 \pm 150$ IgG/s, median IgM secretion rate $300 \pm 117$ IgM/s). Therefore, all HK-B were able to induce IgG-SCs, although at varying frequencies. Sequential to the beadline assay, we performed the bactoline assay to characterize the anti-bacterial surface binding IgG-SCs (Fig. 3b). As expected by their low frequency of IgG-SCs (extracted from the beadline assay, Fig. 4a, b), negative control mice (adjuvant only) and HK-PA immunized mice showed no bacterial-surface binding IgG. In mice immunized with HK-ST and HK-EB, we observed bacteria-specific IgG-SCs with frequencies of $0.68 \pm 0.45\%$ and $0.40 \pm 0.38\%$ (both $n = 3$, Fig. 4c). In comparison with their total IgG-SCs frequencies (3.12% and 2.17%, respectively), we could determine that only around 20% of all secreted IgGs bound to bacterial surface antigens.

Next, we wanted to use the calibration curves shown in Fig. 2f, g to estimate the concentration of available epitopes and $K_{D\,app}$. However, when we performed the cell experiments described herein, the measure along the $x$-axis was time; more specifically, the time at which the image was taken, and not concentration as displayed in Fig. 2f, g. To convert time to concentration (see Fig. 3b, $x$-axis), we approximated the in-droplet antibody concentration by using the median IgG secretion rate ($250 \pm 150$ IgG/s) that was measured in parallel by the beadline assay with the same pool of B cells. Since individual cells secrete different amounts, such an assumption will lead to variances in the extracted $K_{D\,app}$ (see also "Discussion"). Due to this variance, the extracted $K_{D\,app}$s were grouped into high (<10 nM), intermediate (10–50 nM) and low (>50 nM) affinity. The measurement of available epitopes ($y_{max}$) was not affected by the use of the median secretion rate but rather by the time resolution that was used to image the droplets. Therefore, according to the ovalbumin calibration experiment in Fig. 2d, available epitopes were also classified into low (<10 nM ovalbumin equivalent, $y_{max} < 2$), intermediate (10–100 nM ovalbumin equivalent, $2 < y_{max} < 3$) and high availability (>100 nM ovalbumin equivalent, $y_{max} > 3$). These categories were used to generate heat-maps of the anti-bacterial secreted IgG repertoire. Figure 4d shows the summary of the analyzed bacterial surface specific antibody diversity in HK-ST and HK-EB immunized mice (due to the absence of specific IgG, adjuvant only and HK-

PA are not shown in the Figure). In HK-ST and HK-EB immunized mice, the majority of bacteria-binding IgGs recognized epitopes that were present at high concentrations with high apparent affinity (around 50%). Almost all of the remaining 50% of anti-bacterial IgGs were distributed between low and intermediate epitope concentrations, and only very few (<5%) of measured IgGs displayed a $K_{D\,app}$ of >50 nM, showing that the immunization with HK-ST and -EB lead to highly affine anti-bacterial IgGs.

**The secreted anti-bacterial IgG repertoire displayed high cross-reactivity.** In our experiments, most of the IgG-recognizing surface antigens on HK-ST and HK-EB did so by binding to very abundant epitopes (Fig. 4d), making the immunodominant LPS a likely candidate antigen. LPS are strain-specific in their glycosylation, yet many variants also share similarities and conserved epitopes[40–42]. Consequently, we were interested in identifying whether the mice immunized with a specific bacteria species produced more specific or cross-reactive antibodies that might interact with other bacteria. Indeed, cross-reactivity was observed when we performed antigen-specific titer measurements against all used bacteria (Supplementary Fig. 6). Here, we were able to observe measurable antibody responses not only against the HK-B used for immunization, but also cross-reactivity among all the HK-B. Next, we assayed the commercial monoclonal antibodies that we initially used for calibration experiments (Figs. 2, 5a), and measured their binding toward the different HK-B. Indeed, substantial levels of cross-reactivity were observed for the two anti-LPS antibodies (anti- HK-ST, anti- HK-EB). The observed fluorescence relocations were specific for these two antibodies since the bactoline did not simply bound any presented antibody (Fig. 2b) nor did the antibodies bind unrelated, very different surfaces (Fig. 5a HK-SA).

When performing the bactoline assays against bacteria not used for immunizations, mice immunized with HK-EB and HK-ST showed noteworthy frequencies of cross-reactive antibodies (Fig. 5b). In both cases, the frequency of cross-reactive IgGs was around 60% of the frequency of IgG binding the immunogen (Fig. 5b). HK-ST immunized mice further showed cross-reactivity with HK-PA, whereas IgGs from mice immunized with HK-EB

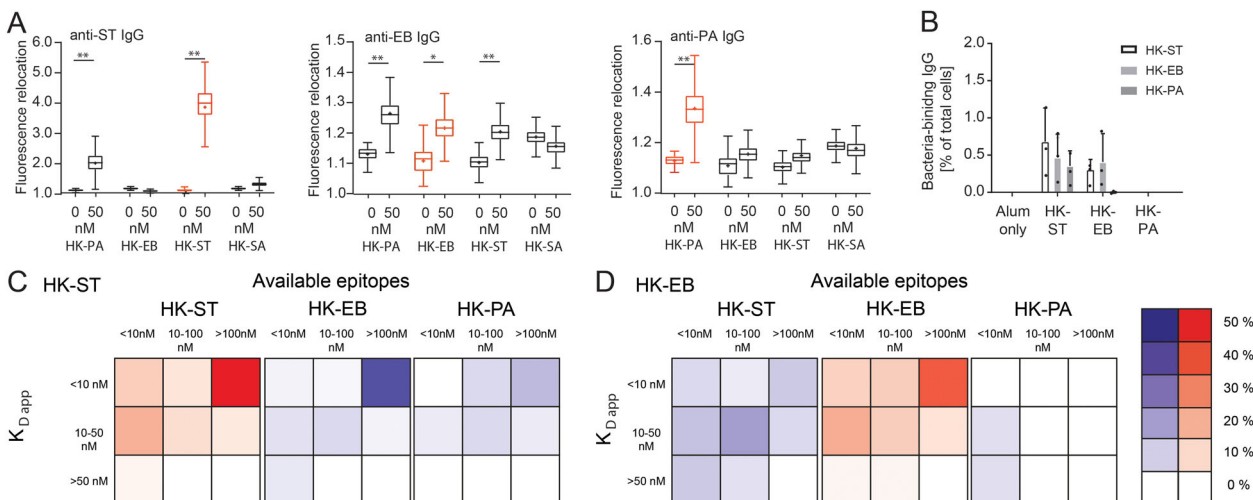

**Fig. 5 Cross-reactivity of commercial and immunization-induced IgGs. a** Cross-reactivity of the used commercial antibodies (Fig. 2); *$p$-value < 0.05; **$p$-value < 0.01; ***$p$-value < 0.001. All data in **a** is shown as Tukey box-and-whisker plots; the corresponding marker depicts the mean relocation ($n > 3000$ droplets for each point). **b** Frequency of cross-reactive antibodies in immunized mice. Data are shown as the mean frequency ± standard deviation over all mice ($n = 3$ per condition). **c** Characterization of cross-reactive IgG in HK-ST immunized, and **d** HK-EB immunized mice. Data are shown as in Fig. 4d (for each panel $200 > n_{anti-HK-B\,IgG-SCs} > 2$). Frequencies of cross-reactive IgGs are shown as a measure of relative frequencies of anti-immunogen IgG-SCs (red, **b**).

did not. Interestingly, when comparing the binding heat-maps of HK-ST immunized mice (Fig. 5c), the map of estimated $K_{D\ app}$ and epitope density was almost identical when the secreted IgG-repertoire was assayed against HK-EB. High-apparent affinity high capacity IgG were found to extensively cross-react with the closely related HK-EB. This indicated that these antibodies recognized epitopes present at high density and which seem similar in *Escherichia* and *Salmonella*, possibly due to their common features in LPS[40–42]. However, and as expected, less cross-reactivity (yet still a considerable amount) was found when these antibodies were assayed against HK-PA, an evolutionarily more distant Gram-negative bacteria (Fig. 5c). The inverse experiment, mice immunized with HK-EB and assayed against HK-ST (Fig. 5d), showed higher specificity of the highly affine antibodies, and the distribution of cross-reactive antibodies was more uniformly distributed over the heat map. Cross-reactivity with HK-PA was almost absent and limited to two antibodies with low apparent affinity and epitope density (Fig. 5d).

**Immunization with a combination of the three HK-Bs resulted in higher frequencies of surface-binding IgG, but lower binding strengths.** Lastly, we were interested in the effect that a combined immunization with all three bacteria would have on the profile of the induced surface specific and cross-reactive IgGs. Therefore, we immunized mice with an equal mixture of the three HK-B cells (PA, EB, ST); in sum $10^8$ HK-B, equal to the same number of bacteria used for immunization with a single bacterial species. The HK-PA + EB + ST-immunized mice (later referred as 3× mice) displayed significantly higher frequencies of total Ig-SCs (10 ± 1.8%) than single bacteria species-immunized mice (p-value for all <0.05). In 3× mice, 5.40 ± 0.38% of Ig-SCs secreted IgG, and 4.79 ± 1.27% secreted IgM, noticeably with an IgG/IgM ratio of around 1.19. Both frequencies were higher than any frequencies measured in single bacterium-immunized mice. Interestingly, the frequency of IgG-SCs in 3× mice was non-significantly different from the sum of the IgG-SCs from the three individually immunized mice (5.92 ± 2.65% and 5.40 ± 0.38%, respectively, p-value = 0.75). Although only a third of each

respective bacteria species was used for immunization in 3× mice, these results revealed a higher immunogenicity of the combination of bacteria.

In consequence, these mice also showed the highest frequency of bacteria-binding IgGs. Within 3× mice, we found frequencies of 1.06 ± 0.39% binding to HK-EB, 1.10 ± 0.50% binding to HK-ST and even 0.66 ± 0.37% binding to HK-PA. It is of interest to note that an immunization with HK-PA alone (3× higher amount of HK-PA) resulted in no PA surface-binding IgGs, and even a thirty-fold higher dose of HK-PA alone resulted in only 0.5% PA-binding IgGs. Therefore, the mixture of HK-PA with different bacteria was able to increase the frequency of HK-PA binding antibodies significantly (0 ± 0.0% to 0.66 ± 0.37%). However, this increase could also be due to the high cross-reactive potential of the antibodies raised against HK-ST. In conclusion, the combination of HK-B for immunization resulted in a mean increase of HK-EB and HK-ST binding IgG by 1.8 ± 0.3-fold, a substantial increase in surface-binding antibodies. This was also observed in titer measurements (Supplementary Fig. 6) in which 3× mice outperformed the single bacteria species-immunized mice.

However, not only were the frequencies of surface-binding anti-bacterial IgG-SCs considerably different, but so were their apparent affinity and epitope density heat maps. When comparing these heat maps between 3x mice and single bacteria species-immunized mice (Fig. 6a, b, for comparison Fig. 5c), the antibodies were more evenly distributed over the full heat map for 3x mice. A second maximum at intermediate affinity/epitope density (middle tile) appeared whereas the high-affinity epitope category showed reduced frequency (46.5 ± 6.5% and 33.5 ± 3.5%, p-value = 0.04). Indeed, the frequencies of highly affine antibodies in 3× mice that recognize epitopes at low or high density were significantly reduced (−9 ± 2% and −13 ± 3%, both p-value <0.05) when compared to individually-immunized mice (Fig. 6c). In total, $K_{D\ app} < 10$ nM antibodies decreased −22% over all available epitopes; and a similar decrease (−19%) was found for antibodies recognizing widely available epitopes. Interestingly, the coherent increase in frequency was found in the intermediate categories, both for affinities and available epitopes (Fig. 6c,

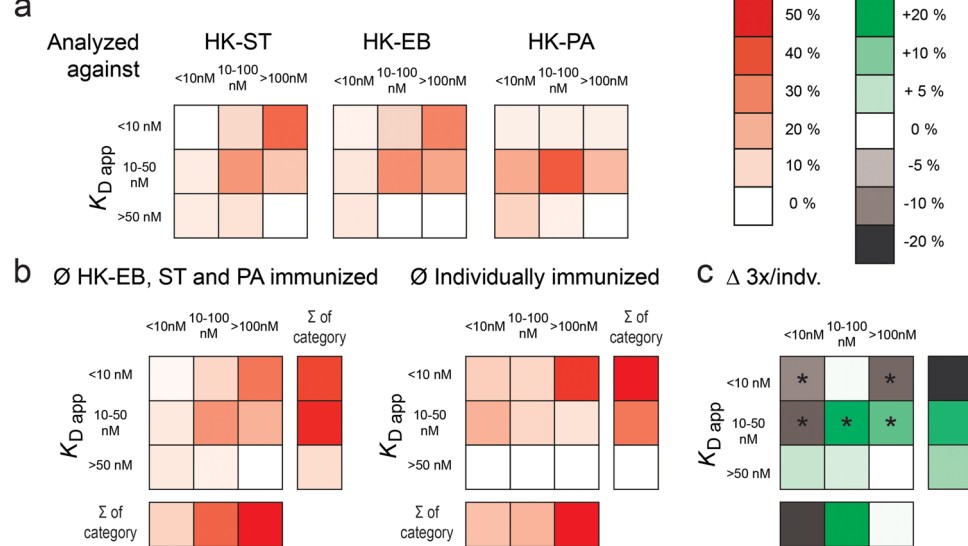

**Fig. 6 Data from mice immunized with all three HK-B. a** Characterization of HK-B-binding IgGs in mice immunized with a mixture of HK-ST, HK-EB, and HK-PA. **b** Average heat-maps found in HK-EB, -ST, and -PA immunized mice, and in individually-immunized mice. Columns to the right and below show the sum of frequency in that category. **c** Difference between frequencies in the various categories found in HK-EB, -ST, and -PA immunized mice and in individually-immunized mice. *p-value < 0.05. Data for all panels is binned as described in the methods; and shown as a relative frequency of IgG in bin over total IgG for all mice in the respective category.

$p$-value for both <0.05). Nevertheless, >30% of IgG-SCs produced IgG that interacted with the epitopes of the highest density on the HK-ST and HK-EB surface (">100 nM") with high apparent affinity in 3× mice.

## Discussion

In this report, we outlined an analytical strategy to describe the anti-bacterial IgG repertoire with single-antibody resolution and high-throughput in a quantitative manner. The developed bioassay is based on the immobilization of heat-killed bacteria on magnetic nanoparticles, forming a solid, stable, and equal surface to perform immunofluorescence assays within each 50 pL droplets. Doing so allowed for an estimation of important biophysical parameters such as $K_{D\ app}$ and epitope density for every secreted anti-bacterial antibody individually. Due to the direct binding of the antibody on a highly repetitive surface (bacteria), mono- or multivalent interactions are possible, and an apparent dissociation constant ($K_{D\ app}$) was extracted as a measure of overall binding strength. However, this interaction is identical with the binding of said antibody to the bacterium in solution, and $K_{D\ app}$ is therefore of functional relevance. In comparison with existing microfluidic approaches, such as described by Shembekar et al. and Gerard et al.[29,30], single-cell heterogeneity of the analyte cells[33–35] was removed due to the presence of multiple bacteria per droplet. Therefore, a much better standardization and quantitative characterization was achieved[28,43]. In comparison to existing technologies such as ELISpot, the herein described workflow allowed to not only measure the frequency of such events, but also provided an estimation of affinity and epitope density with single-antibody resolution.

Certain limitations restrict the broad application of the developed strategy and the immunological conclusions, limitations that need to be addressed in subsequent studies. First, the recognized epitope and antigen remained unknown in the described assay format. Most of the identified antibodies recognized antigens of high capacity, present at high concentration in the bacterial membrane. Given our employed immunization with full HK-B, it is reasonable to assume that immuno-dominant antigens, such as LPS, are favored antigens for these antibodies. Given the presence of conserved epitopes on LPS, this hypothesis would also explain the extensive cross-reactivity found within the samples. Further experiments are needed to identify the epitopes of the generated antibody repertoire before additional conclusions can be drawn from the data, and added experiments need to clarify and quantify the role of T-cell help in the observed cross-reactivity of the secreted IgGs. Second, the use of heat-killed variants for immunization and the bioassay itself poses an additional issue due to potential alteration of protein epitopes. The influence of heat killing on epitope structure and presence was not accessible in our studies but must be evaluated when the system is used to find potential therapeutic antibodies against protein antigens from the bacterial membrane. Lastly, the experiments solely provided an estimation of $K_{D\ app}$. For calibration experiments and modeling, we used known and controlled antibody concentrations, whereas the in-droplet concentration in a cell experiment was unknown due to the unknown secretion rate of the corresponding cell, and the present antibody concentration was estimated by using the median secretion rate. Since individual cells secrete different amounts, such an assumption will lead to variances in the estimation of $K_{D\ app}$. Per definition, 68% of cells will secrete within 1 standard deviation, and calculations of the extracted $K_{D\ app}$ was found to vary by a factor of 3× when a median value ± 1σ was used for calculation. Although this deviation was found to be larger than with other methods such as SPR (with a coefficient of variation

between 0.1 and 0.4), this estimation still represented a good approximation of $K_{D\ app}$ on the single-cell/antibody level.

Independent of these limitations, the experiments still reveal that identical immunizations with different HK-B can result in vastly different IgG responses. HK-ST and HK-EB immunized mice showed high and moderate immunogenicity, respectively, while HK-PA showed very weak immunogenicity. Interestingly, an immunization with a combination of all three bacteria outperformed individual immunizations in almost all parameters, although the relative frequency of high-affinity antibodies ($K_{D\ app}$ < 10 nM) was reduced. Instead, a substantial shift toward intermediate affinity and epitopes density was found in these mice. Likewise, titer responses of the 3x mice outperformed the individually-immunized mice, and even a good titer response against HK-PA was found that was absent in HK-PA-only immunized mice (see also the Supplementary Information). However, it is not clear if these antibodies bind HK-PA due to cross-reactivity or if the addition of the other HK-B enabled the immune system to recognize and process HK-PA. While we were able to visualize the reported, low affinity, poly-specific antibodies, we also found that high-affinity IgGs were highly cross-reactive between different, but closely related bacteria. Although anti-bacterial antibody poly-specificity has been described before[20,21], it will be of further interest to use this system to study cross- and poly-reactivity of anti-bacterial antibodies with regards of vaccination, the microbiota and autoimmune diseases in a quantitative manner[44]. Especially in the context of autoimmune diseases, assaying cross- and poly-reactivity of the secreted antibody repertoire is of great fundamental interest to advance the understanding of the pathophysiological mechanisms and genesis of these diseases.

A thorough characterization of antibody repertoires will also benefit screenings for antibodies used in research, diagnosis, or therapy[45], and may help to identify candidates of interest. Whilst the isotype of a therapeutic antibody can be adapted to induce specific functionalities[5,6], the affinity and especially the density of the recognized epitope on the bacterium cannot be altered easily for a specific antibody. Therefore, a thorough characterization of antibodies and their epitopes before sequencing can leverage the laborious steps of antibody expression and characterization. Here, anti-bacterial antibodies are a valid strategy to counter the negative impact of antibiotic resistance on individual and global health[46,47], and the introduced bioassay has the potential to screen for such anti-bacterial antibodies. Our bioassay allowed us to measure antibodies against both Gram-positive and -negative bacteria, and the immobilization strategy can be expanded to a variety of different membrane antigens found in membranes of eukaryotic cells or viruses[32]. The assay's potential to provide an all-in-one functional characterization prior to downstream screening and sequencing is of great interest[29,30]. Given the ubiquity of antibodies in research today, a system that streamlines the characterization of antibodies in terms of affinity, specificity/cross-reactivity[48–50] and epitope density can prove essential.

## Methods

**Immunization of mice.** 8–10 weeks old BALB-C Mice (Janvier Labs, female) were immunized i.p. with $10^8$ HK-B (HKST, HKPA, HKEB2, all Invivogen). For immunization, HK-B were added to 0.9% (w/v) NaCl (Versylene Fresenius) and mixed 1:1 with alhdyrogel adjuvant 2% (Invivogen). For the negative control mice, the mice were immunized with NaCl and adjuvant alone. After 6 weeks, the animals were boosted using the same bacteria in adjuvant, and 6 days thereafter the animals were sacrificed and the spleen was extracted. Each immunization was performed in three individual mice ($n = 3$). Cells from the B-cell lineage were isolated as described in ref.[31] using a commercial kit (Miltenyi Biotec, Pan B Cell kit II). Here, the cells were incubated for 5 min in Red blood cell lysis buffer (BD) instead of ammonium chloride buffer (decreased stress, increased Ig secretion rates, own observations). Experiments using mice were validated by the CETEA ethics

committee number 89 (Institute Pasteur, Paris, France) under #2013-0103, and by the French Ministry of Research under agreement #00513.02.

**Aqueous phase I: preparation of cells or control antibodies solution**. Aqueous phase I contained either the isolated cells (cellular measurements) or purified monoclonal antibodies (calibration experiments). For the list of calibration antibodies used, please refer to Supplementary Table 1. The isolated cells were centrifuged at $300 \times g$ for 5 min, then the cell pellet was gently re-suspended in cell medium to achieve a concentration of 5–10 million cells/ml. In order to minimize antibody secretion before droplet formation, the cell suspension was prepared just before droplet formation; and kept on ice for short-term storage. Cell medium was composed of RPMI 1640 with no phenol red, supplemented with 0.1% Pluronic F-68, 25 mM HEPES pH 7.4, 10% KnockOut Serum Replacement (all ThermoFisher), and 0.5% recombinant human serum albumin (Sigma Aldrich, A0237).

**Aqueous phase II: preparation of nanoparticles and detection antibodies solution**. Aqueous phase II contained streptavidin-coated paramagnetic nano-particles (Streptavidin plus, diameter 300 nm, Ademtech) that are either coated with biotin anti-kappa light chain $V_H H$ as described in ref. [31,51] (termed beadline); biotinylated ovalbumin (labeled in-house, Invivogen) or heat-killed bacteria (HK-B, termed bactoline, see also Figs. 1, 2). For bactolines, the HK-B (all Invivogen) were first thoroughly re-suspended and individualized by a 30-time passage through a 34 G needle (Terumo). The removal of clusters was visually confirmed by microscopy; and by a subsequent increase in $OD_{600}$ measures (up to 2.5-fold increase for all assayed HK-B). The individualized HK-B were centrifuged ($6000 \times g$, 2 min); and the pellet was re-suspended in 10 µl of PBS containing 100 µM Biotin-PEG-Cholesterol (PG2-BNCS-3k, Nanocs)[32]. The mixture was incubated overnight at 4 °C to allow the integration of the anchor molecule. Afterward, the HK-B were washed by centrifugation ($6000 \times g$, 2 min), followed by re-suspension in PBS containing the magnetic nanoparticles. Per 25 µl of nanoparticles stock solution, we used $4 \times 10^9$ HK-B. The mixture was incubated for 2–3 h whilst gentle shaking at RT. Afterward, the nanoparticles were collected using a magnet; and washed to remove unbound HK-B. The nanoparticles were then re-suspended and incubated in cell medium for 30 min at RT to block non-specific binding, washed again and re-suspended in 50 µl cell medium ($2\times$ volume of the initial bead stock solution). Afterward, anti-IgG Fc Alexa647 (Jackson ImmunoResearch, cat.no. 315-606-046) and anti-IgM µ-chain Alexa555 (Tebu Bio, 221W99020C) were added to achieve a final in droplet concentration of 45 nM for each labeled antibody. For ovalbumin experiments, the beadlines were directly covered with different amounts of biotinylated ovalbumin (DOL 2) instead of $V_H H$. Afterward, the OVA-covered nanoparticles were further processed as described in ref. [31,51].

**Droplet formation and chamber filling**. Water-in-oil emulsion droplets were generated using a custom-made microfluidic chip as described in ref. [28,51]. The droplets were transferred from the chip outlet to the 2D observation chamber (as described in ref. [51]) via a microtube (Adtech Polymer Engineering™, PTFE, 0.3 mm inner diameter), passing through a ring magnet (H250H-DM, Amazing Magnets). Once the chamber was filled with the droplets, the chamber was closed and mounted onto a fluorescence microscope for imaging.

**Imaging and data acquisition and analysis**. The droplets in the 2D chamber were imaged using an inverted fluorescence microscope (Eclipse Ti-2, Nikon) equipped with a motorized stage and excitation light (Lumencor Spectra X). The images were recorded using a digital CMOS camera (ORCA-flash4.0 V2, Hamamatsu, in $2 \times 2$ binning mode) at an exposure time of 100 ms through a $10\times$ objective (NA 0.45; Nikon). Each cell experiment was acquired as an array of $10 \times 10$ images, which covered around 30,000 50 pL droplets. Images were recorded every 15 min for an hour (five time point measurements total) using appropriate FITC, TRITC and Cy5 filters. The images were analyzed by a custom Matlab script as described in refs. [31,51]. To measure bactoline intensity, the algorithm selected the brightest pixels (mean area of bactoline) in the droplet within the FITC channel (auto-fluorescence of HK-B). Smaller isolated fragments, less than half of the total area, were removed from analysis (removal of free-floating HK-B, noise). The output, a binary mask corresponding to the bactoline area, was used to measure fluorescence relocation within the other channels. Droplets containing Ig-SCs were identified by a detectable increase in fluorescence relocation and the presence of a cell. IgG secreting cells were detected by anti-IgG Fc specific secondary antibody conjugated to Alexa647 (red fluorescence), and IgM secreting cells were detected by anti-IgM mu specific secondary antibody conjugated to Alexa555 (yellow fluorescence). Beadline data was analyzed as described in ref. [31,51]; example data and access to the software can be found through ref. [51]. The total number of screened cells was calculated by multiplying the total number of droplets by the λ (cells/droplets). Secretion rates were calculated as describes in reference[31].

**ELISA protocol**. First, 100 µl of PBS were added to each well, and afterward 2.5 or $5 \times 10^8$ of HK-B were added (in 50 µl PBS) into each well of a 96-well Maxisorp Nunc-Immuno plate (Thermo Scientific). The plates were incubated at 4 °C over-night; and the next day the wells were thoroughly washed using PBS. The wells were

afterwards blocked using a solution of 0.5% recombinant human albumin (rHSA, Sigma-Aldrich, A9731) in PBS for two hours at RT. Subsequently, the plates were washed two times with PBS. 100 µl of serial dilutions of serum or antibodies at indicated concentrations were added to the wells; and the plates were incubated for 2 hours at RT (or overnight at 4 °C). As diluent, 0.1% rHSA in PBS was used. After incubation, the plates were washed twice using 0.1% rHSA. For detection, we used goat anti-Mouse IgG horseradish peroxidase (HRP) conjugate (Invitrogen, A16166) at a final concentration of 50 ng/ml in PBS with added 0.1% rHSA. Of this solution, 100 µl were added for two hours at RT. Afterward, the wells were thoroughly washed three times with 200 µl PBS. To develop the signal, 100 µl of a freshly prepared solution containing 0.1% rHSA, 100 µM hydrogen peroxide (SigmaAldrich) and 5 µM Amplex red (ThermoFisher) were added to the wells, and the plate was immediately read in a Tecan Infinite 200 Pro (Tecan, excitation 535 nm, emission 590 nm) in a kinetic mode (every 10 min for 1 h). Since the resulting data were found to be linear in the observed time range, the slope over time was calculated for data analysis and compared. Alternatively, the signal intensity at a specific time was used.

**Estimation of binding strength and epitope availability using ELISA**. As for the estimation of the binding strength (apparent dissociation constant ($K_{D\ app}$), we used a method as published by Beatty et al.[52]. The data from the two different antigen/epitope concentrations were fitted individually using a Hill equation; and the respective $EC_{50}$ values were extracted. According to Beatty et al.[52], the $K_{D\ app}$ can be estimated using the following formula (1):

$$K_{D\ app} = \frac{1}{2(2 \times EC50_{1x} - EC50_{2x})} \quad (1)$$

with $EC50_{1x}$ being the extracted EC50 for lower epitope concentration, and $EC50_{2x}$ for the double epitope concentration. Using the same fitting curves, the epitope density was estimated by extracting the maximal enzymatic turnover rate for the commercial antibodies. This maximal turnover rate corresponded with the plateau value of the Hill curve at high antibody concentration. This plateau value referred to the highest amount of antibody that was bound to the bacteria, and therefore correlated with the amount of binding sites present within a well (see also Fig. 2).

**Estimation of apparent dissociation constant ($K_{D\ app}$) and the concentration of available epitopes from cell measurements**. Due to restrictions in the calculations of the assay, only droplets with a defined and observed maximum in fluorescence relocation were processed for calculations (see also Supplementary Information). For the estimation of $K_{D\ app}$, the extracted maximal fluorescence relocation ($y_{max}$) was corrected by the concentration that was present within the droplet when this maximum was reached:

$$a = \frac{y_{max}}{(x_{max} \times \overline{SR} \times corr)} \quad (2)$$

with $y_{max}$ being the maximal fluorescence relocation value observed, $x_{max}$ the time at which this maximum was observed, SR as the median secretion rate (see also Supplementary Information) and corr as a correction factor for droplet volume. Using the calibration curve (Fig. 2), the $K_{D\ app}$ was calculated as:

$$K_{D\ app}[nM] = e^{\frac{\alpha - 0.969}{-0.014}} \quad (3)$$

Due to the use of a median secretion rate for the estimation of $K_{D\ app}$, the extracted $K_{D\ app}$ were grouped into high (<10 nM), intermediate (10–50 nM) and low (>50 nM) $K_{D\ app}$. To further estimate the number of available epitopes that the antibody binds to, the curve shown in Fig. 2 was used. The present concentration of epitopes for the bacterial antibodies to bind to were classified using the respective $y_{max}$ values. Available epitope concentration was classified into low (<10 nM ovalbumin equivalent, $y_{max} < 2$), intermediate (10–100 nM ovalbumin equivalent, $2 < y_{max} < 3$) and high availability (>100 nM ovalbumin equivalent, $y_{max} > 3$).

**Statistics and reproducibility**. All data in this work were collected in multiple ($n = 3$) and respective data are either means ± SD (frequencies) or medians ± SD (secretion rates) of replications over individual mice. In the heat-map figures, affinity and epitope density data was presented cumulative for all three mice together. Statistical analyses for significance shown in figures and text were calculated using two-tailed Students $t$-tests.

**Reporting summary**. Further information on research design is available in the Nature Research Reporting Summary linked to this article.

## Data availability

The original data used in this publication are made available in a curated data archive at ETH Zurich[53] (https://www.research-collection.ethz.ch/handle/20.500.11850/431634) under the https://doi.org/10.3929/ethz-b-000431634.

## Code availability

The custom DropMap Matlab script is available from a GitHub repository under https://github.com/LCMD-ESPCI/dropmap-analyzer.

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

## Acknowledgements

This work was supported by the CELLIGO project funded by the French government through BPIFrance under the frame "Programme d'Investissements d'Avenir" (PIA), the "Institut Pierre-Gilles de Gennes" through the laboratoire d'excellence, "Investissements d'avenir" programs ANR-10-IDEX-0001-02 PSL, ANR-10- EQPX-34 and ANR-10-LABX-31, and the European Research Council (ERC)–Seventh Framework Program

(ERC-2013-CoG 616050 to P.B.). E.K. acknowledges generous funding from the 'The Branco Weiss Fellowship—Society in Science' and received funding from the European Research Council (ERC) under the European Union's Horizon 2020 research and innovation programme (Grant agreement No. 80336). C.C. acknowledges financial support from CONCYTEC, Peru.

## Author contributions

M.H., G.C., C.C. and K.E performed and optimized the experiments described in this protocol and G.C. provided the respective Matlab Script for data analysis. K.E., J.Bi., J.Ba., and A.D.G. developed the underlying DropMap technology. J.Bi., A.D.G., J. Ba., P.B., and K.E. supervised the work and M.H., J. Ba. and K.E. designed the experiments. M.H., G.C., and K.E. analyzed the data, and M.H., C.C., J.Ba., P.B., and K.E. wrote the paper. All authors edited and proofread the paper.

## Competing interests

The authors declare the following competing interests: J.Ba., J.Bi., A.D.G., and K.E. are inventors on patent applications based on certain ideas described in this manuscript and may receive financial compensation via their employer's rewards to inventors scheme. M.H., G.C., C.C., and P.B. declare no competing interests.
