## [Peer Review File · Communications Biology]

Reviewers' comments:

Reviewer #1 (Remarks to the Author):

Heo and Eyer et al in the manuscript entitled "Deep-phenotypic characterization of the immunization induced anti-bacterial immunoglobulin-G repertoire" develop a novel assay platform to evaluate antibody specificity, cross reactivity and affinity in a high throughput manner at single molecule level. The manuscript is written well and advances the field by potentially developing a method that can be broadly applicable to diverse antibody dependent basic and translational research avenues. Therapeutic antibodies development can be greatly advanced if this technique is broadened for use to other eukaryotic antigens/virus, cancer surface antigens etc.

The following points if addressed would make the manuscript more compelling:

1. Can the authors compare immunization of HK-B with LPS purified from E.coli and LPS purified from Salmonella (and a purified antigen from Gram +ve bacteria)? It is likely that LPS in Gram -ve bacteria is an immunodominant antigen. It would be interesting if response to an immunodominant antigen would interfere or compete with antibodies generated from a diverse polyclonal response against total HK bacteria
2. Can the authors discuss about antibody responses generated via T-dependent and T-independent manner? NP-LPS or NP-Ficoll primarily generate TI-I or TI-II responses. Antibody affinity is greatly affected by Tcell help. Experiments in T cell deficient mice might be important to address if antibody affinity and cross reactivity would be influenced if the response becomes more tailored with time due to evolution of antibody specificity in GC by Tc help
3. Can the authors discuss or address experimentally differences that might be due the process of "heat killing". Surface antigens (especially lipoproteins in membrane for eg.) in natural conformation might invoke different antibody response and affinity/cross reactivity might vary from what authors see for HK-B

Minor comments:

1. Gram "G" capitalize
2. typo and grammar incorrect line 79, 313 of pdf version.
3. Elaborate discussion to include aspects of how this could be potentially used in Immunology, autoimmunity therapeutic/translational research

Reviewer #2 (Remarks to the Author):

The manuscript by Heo et al describes an assay to measure antibody responses to bacterial antigens after experimental immunization. The assay makes use of heat-killed bacteria immobilized to nanoparticles as antigen. Several characterization steps have been made to optimize and calibrate the assay, in order to extract information about antibody affinity and density of recognized epitopes.

In general, I found the manuscript very difficult to read. The figures are not self-explanatory and the results make constant reference to information in the methods section, which is disruptive, primarily because this is a methods paper. It is unclear what is the advantage of this assay to commonly used approaches such as ELISAS, ELISOT or Biocore, normally used to measure content, affinity or frequency of Ab secreting cells.

Specific points

- 1) The assay uses anti-IgG secondary antibodies recognizing a range of polyclonal anti-bacteria antibodies. This information is used to infer different parameters including antibody affinity, etc. How are the authors controlling for the diverse IgG isotypes that can be present in a complex antibody mixture, specially after immunization that can be influencing the binding of the conjugate to the Ag-specific antibody?
- 2) In Figure 2, the authors refer to calculations of antibody affinity. There is no information of

whether the antibodies used are monoclonal or polyclonal. Affinity can be estimated only through the binding of one monoclonal antibody to its epitope. Anything else, should be referred as avidity estimations.

3) It is unclear how maximal ELISA can be used to estimate the number of available antigens.

4) Figure 2F is not described in the results text.

5) In general, it is unclear what is the information that this assay can provide that cannot be deduced from established antibody assays.

Reviewer #3 (Remarks to the Author):

This paper uses a novel nanoparticle technique to examine the individual antibody repertoire to a number of bacterial species in mice. The paper shows that immunisation with individual bacteria induce a variable antibody secreting cell number, ranging from 3.8-6.4% of total ASCs. In addition, the authors show that 60% of the bacteria-specific IgG is cross-reactive, an interesting finding that is at odds with current taught immunology on the antibody response. Another very interesting finding is that immunizing with multiple bacterial species, as opposed to single bacterial immunisations, reduces both the frequency and affinity of ASCs.

I have very few issues with this manuscript. It describes a series of validation experiments for the new technology, and performs some very simple and interesting tests in mice. The assay has the potential to be scaled up and, perhaps most importantly, would be of great value in targeting human populations to examine ASC responses following vaccination. The Discussion is thorough and fair, and does not oversell the findings of the paper. Overall, a very nice study.

Only two minor issues that the authors should address in their manuscript:

1. The authors reference Figure 1 and S1 Figure 1 in the Introduction. Results should not appear in the Introduction and this reference should be removed.

2. In Figures 1 and 2, why does the fluorescence relocation drop off at higher concentrations? I would have thought it should simply plateau at maximal antibody concentration? I think a brief explanation would certainly help the non-specialist understand this aspect better.

Reviewers' comments:

Reviewer #1:

Can the authors compare immunization of HK-B with LPS purified from E.coli and LPS purified from Salmonella (and a purified antigen from Gram +ve bacteria)? It is likely that LPS in Gram -ve bacteria is an immunodominant antigen. It would be interesting if response to an immunodominant antigen would interfere or compete with antibodies generated from a diverse polyclonal response against total HK bacteria	Due to the current pandemic crisis, we were not able to perform these experiments in the given timeframe – we have discussed this issue with the editor of the journal. We had immunized the mice, but currently do not have access to them due to the lockdown in France. In the current situation, it is also difficult to plan these experiments in the near future. We are sorry for this. Instead, we have added a part in the discussion that debates this question. However, we agree with the reviewer and the editor that this question is of great interest, and we will perform these experiments in a later study that will look at this scientific hypothesis in much more detail; also with in regard with the second comment of reviewer 1.
Can the authors discuss about antibody responses generated via T-dependent and T-independent manner? NP-LPS or NP-Ficoll primarily generate TI-I or TI-II responses. Antibody affinity is greatly affected by Tcell help. Experiments in T cell deficient mice might be important to address if antibody affinity and cross reactivity would be influenced if the response becomes more tailored with time due to evolution of antibody specificity in GC by Tc help	We have added and modified a specific paragraph in the discussion. We had planned some experiments to study this in more detail, but due to the current lockdown, these studies had to be postponed.
Can the authors discuss or address experimentally differences that might be due the process of "heat killing". Surface antigens (especially lipoproteins in membrane for eg.) in natural conformation might invoke different antibody response and affinity/cross reactivity might vary from what authors see for HK-B	We have added and modified a specific paragraph in the discussion.
Gram "G" capitalize	Thank you very much for pointing this mistake out. We have corrected it throughout the manuscript.
typo and grammar incorrect line 79, 313 of pdf version.	Both sentences in question have been corrected and re-written.
Elaborate discussion to include aspects of how this could be potentially used in Immuno oncology, autoimmunity therapeutic/translational research	We have added and modified a specific paragraph in the discussion.

Reviewer #2

In general, I found the manuscript very difficult to read. The figures are not self-explanatory and the results make constant reference to information in the methods section, which is disruptive, primarily because this is a methods paper.	We are sorry for the difficulty. We have revised the text to make it clearer, and rearranged and included a novel version of Figure 1 and 2 to help the reader along the manuscript. In addition, we have removed some of the unnecessary references to the method section throughout the manuscript to ease the read of the text. We further tried to better structure the text and remove/introduce information to ease the reading of the text.
The assay uses anti-IgG secondary antibodies recognizing a range of polyclonal anti-bacteria antibodies. This information is used to infer different parameters including antibody affinity, etc. How are the authors controlling for the diverse IgG isotypes that can be present in a complex antibody mixture, specially after immunization that can be influencing the binding of the conjugate to the Ag-specific antibody?	The used secondary reagent has been already described in our original publication, Eyer et al., Nature Biotechnology, 2017, to be able to bind IgG1 and both subtypes of IgG2 (IgG3 is only present in a very small frequency). We have added the following sentence to the manuscript in addition with the references: “The employed fluorescence detection assay for IgG was able to detect IgG1, IgG2a and 2b, whereas the IgM reporter was specific for IgM.”
In Figure 2, the authors refer to calculations of antibody affinity. There is no information of whether the antibodies used are monoclonal or polyclonal. Affinity can be estimated only through the binding of one monclonal antibody to its epitope. Anything else, should be referred as avidity estimations.	All of the measured antibodies were monoclonal antibodies, both the reagents to set up the assay as well as afterwards for measuring cell secretion. We have added this remark when we discuss the results and mention this in the caption of Figure 2.
It is unclear how maximal ELISA can be used to estimate the number of available antigens.	The estimation of the available antigens is not performed using ELISA. This might have been also unclear from our description. We updated the text. Instead, we directly calculated/estimated the number of available antigens from the measurements performed within the in-droplet assays (Figure 2D and y-axis Figure 2F); where we used different amounts of immobilized OVA on the beadlines, therefore also their units as nM OVA equivalent. The maximum turnover rate in ELISA was correlated with y_{max} as a control experiment since the number of antigens/epitopes available in a direct ELISA is directly correlated with its ability to bind antibody. We have added more explanation to the text; an modified Figure 2 and its caption:

	'To conclude, y_{max} was consequently used to determine the in-droplet concentration of available epitopes; and the y_{max} values extracted from Figure 2B were used to define the categories of low (<10 nM OVA equivalent, $y_{max} < 2$), intermediate (10-100 nM OVA equivalent, $2 < y_{max} < 4$) and high antigen availability (> 100 nM OVA equivalent, $y_{max} > 4$).'
Figure 2F is not described in the results text.	We have added the following paragraph to discuss the panel: 'Lastly, we assayed the assays capability to be used with primary cells, and to detect antigen-specific antibodies. Here, we extracted splenocytes from OVA-immunized mice and measured the ability of the secreted antibodies to bind to the antigen that was immobilized on the beadlines. Indeed, the developed direct fluorescence relocation assay was able to visualize antibody binding to the antigen when primary cells were used (Figure 2H).'
It is unclear what is the advantage of this assay to commonly used approaches such as ELISAS, ELISOT or Biocore, normally used to measure content, affinity or frequency of Ab secreting cells. In general, it is unclear what is the information that this assay can provide that cannot be deducted from established antibody assays.	We have added and modified a specific paragraph in the discussion. To the best of our knowledge, biacore and ELISA have not been used in this context with single-cell resolution.

Reviewer #3:

The authors reference Figure 1 and S1 Figure 1 in the Introduction. Results should not appear in the Introduction and this reference should be removed.	The references to the figures have been moved to the result section.
In Figures 1 and 2, why does the fluorescence relocation drop off at higher concentrations? I would have thought it should simply plateau at maximal antibody concentration? I think a brief explanation would certainly help the non-specialist understand this aspect better.	This is due to the Hook effect that is often present in homogenous immunoassays. In our specific case, once the capacity of the beadline is exhausted, any free antibody will compete with the immobilized antibody for the detection antibody, therefore removing detection antibody from the beads. We have added references for this effect in the manuscript, and the following sentence: 'The shape of the curves, increasing fluorescence relocation followed by a decrease, are typical for homogenous immunoassays displaying a Hook effect^{[38, 39][28]}. Above the capacity of the beadline for the antibody, any additional molecules will compete with the immobilized antibodies for the detection antibodies, therefore resulting in a decrease in fluorescence relocation.'

REVIEWERS' COMMENTS:

Reviewer #1 (Remarks to the Author):

I can understand the logistical and resource limitations imposed by COVID-19 in France particularly. It would have been good if authors could experimentally address at least one or two comments from reviewers. The discussion can be expanded to incorporate a section on caveats/limitations of present study, with clear communication of why and how such limitations need to be addressed.

Reviewer #3 (Remarks to the Author):

The authors have responded adequately to all reviewer comments in my opinion, and have amended the text as per each reviewer request.